# Efficient Selective Removal of Radionuclides by Sorption and Catalytic Reduction Using Nanomaterials

**DOI:** 10.3390/nano12091443

**Published:** 2022-04-23

**Authors:** Min Xu, Yawen Cai, Guohe Chen, Bingfeng Li, Zhongshan Chen, Baowei Hu, Xiangke Wang

**Affiliations:** 1School of Life Science, Shaoxing University, Huancheng West Road 508, Shaoxing 312000, China; rocketxm11@163.com (M.X.); caiyawen1993@163.com (Y.C.); hbw@usx.edu.cn (B.H.); 2Zhejiang Zhongguangheng Testing Technology Co., Ltd., Shaoxing 311899, China; 3Power China Sichuan Electric Power Engineering Co., Ltd., Chengdu 610041, China; libing7826450@163.com; 4College of Environmental Science and Engineering, North China Electric Power University, Beijing 102206, China; zschen@ncepu.edu.cn

**Keywords:** radionuclides, sorption, catalytic reduction, nanomaterials, elimination

## Abstract

With the fast development of industry and nuclear energy, large amounts of different radionuclides are inevitably released into the environment. The efficient solidification or elimination of radionuclides is thereby crucial to environmental pollution and human health because of the radioactive hazardous of long-lived radionuclides. The properties of negatively or positively charged radionuclides are quite different, which informs the difficulty of simultaneous elimination of the radionuclides. Herein, we summarized recent works about the selective sorption or catalytic reduction of target radionuclides using different kinds of nanomaterials, such as carbon-based nanomaterials, metal–organic frameworks, and covalent organic frameworks, and their interaction mechanisms are discussed in detail on the basis of batch sorption results, spectroscopy analysis and computational calculations. The sorption-photocatalytic/electrocatalytic reduction of radionuclides from high valent to low valent is an efficient strategy for in situ solidification/immobilization of radionuclides. The special functional groups for the high complexation of target radionuclides and the controlled structures of nanomaterials can selectively bind radionuclides from complicated systems. The challenges and future perspective are finally described, summarized, and discussed.

## 1. Introduction

Nuclear energy is one of the most important backbones of the energy supply, providing ~11% of electricity globally, and ~2% in China [1]. With the rapid development of the nuclear industry and nuclear energy, large amounts of radionuclides are released into the environment during the mining process, the retirement of nuclear power plants, nuclear waste treatment, and even during nuclear accidents. Long-lived radionuclides, which have high radioactivity in the natural environment, can lead to environmental and health problems throughout the food chain. Thereby, the efficient removal or recovery of radionuclides from large volumes of wastewater or seawater is highly important for nuclear power utilization, not only with respect to environmental pollution treatment, but also with regard to the sustainability of nuclear energy [2,3,4]. Different kinds of methods, such as adsorption, solvent extraction, membrane filtration, (co)precipitation, reduction–solidification, ion exchange, etc., have been applied for efficient removal/extraction of radionuclides from large volumes of aqueous solutions [5,6,7]. Among the abovementioned techniques, adsorption has been widely applied in the removal of radionuclides because of its advantages, such as high adsorption capacity, low cost, easy operation, high selectivity, and large scale applicability, in real applications [8,9].

Among the radionuclides, uranium (U(VI)) is one of the most important radionuclides, because it is widely used in nuclear power. The release of U(VI) into the natural environment during the nuclear energy utilization processes is inevitable. Once digested or inhaled, U(VI) will lead to irreversible human health problems. The treatment of wastewater containing U(VI) is a global challenge because of its high radioactivity and coexistence with other metal ions that compete with U(VI) for binding to adsorbents [10,11]. Technetium-99 is one a representative anion that is one of the most important fission products from nuclear fuel processes. ^99^Tc is the fission product of U-235 and/or Pu-239 at very high yield efficiencies of about 6.1% in nuclear power plants and nuclear fuel processes. The long half-life (T_1/2_ = 2.13 × 10^5^ a), high mobility, and remarkable radiation activity give it a higher toxicity, making it of greater environmental concern than other radionuclides [1]. Since the first nuclear reactor, about 400 metric tons of ^99^Tc have been produced [12]. ^99^Tc primarily exists as +7, the most stable valence state (i.e., the pertechnetate anion, ^99^TcO_4_^-^). The non-complexation properties of ^99^TcO_4_^-^ result in high solubility and mobility in the natural environment. Therefore, it is urgent to eliminate ^99^TcO_4_^-^ from radioactive wastewater, not only for purposes of nuclear waste treatment, but also for environmental protection. As these two radioactive elements are highly important for nuclear energy applications, it is urgent to research methods for the removal/extraction of U(VI) and Tc(VII) from wastewater and seawater, not only for environmental protection, but also to ensure the supply of nuclear energy. In this review, recent works focusing on the preconcentration, reduction and solidification of U(VI) and Tc(VII) from large volumes of solutions using nanomaterials are summarized and discussed.

Nanomaterials, such as carbon-based nanomaterials, covalent organic frameworks (COFs), and metal–organic frameworks (MOFs), have attracted multidisciplinary interest because of their special properties, which include low density, high specific surface area, good chemical stability, controlled pore structure, and surface functional group design [13,14,15]. These kinds of nanomaterials have been extensively applied in environmental pollution management. Wang and Zhuang [13] reviewed the application of COFs for the adsorption of heavy metal ions and radionuclides, and photocatalytic/electrocatalytic reduction of heavy metal ions and radionuclides. The COFs showed good sorption ability towards many radionuclides, persistent organic pollutants, and heavy metal ions from wastewater. The abundant porous channels and high specific surface areas are favorable for the quick diffusion of radionuclide ions from the surface to the inner channels. The nanosized pore structure could provide selective passages for the selective separation of pollutant ions or molecules. The incorporation of COFs with membranes can also change the mechanical strength and hydrophilic/hydrophobic properties, thereby enabling the separation of target pollutants from complicated solutions [16]. Most of the reviews about COFs are mainly related to their design concepts, synthesis, structural study, and applications in energy storage, semiconductor devices, drug delivery, and sensors [17,18,19,20]. MOFs are crystalline porous materials that consist of metal nodes and organic linkers. The highly ordered framework of porous structures, permanent porosities, high specific surface areas, and organic linkers such as sulfonates, carboxylates, amines and phosphates give them remarkable sorption ability for metal ions and radionuclides. In our previous review [21], we summarized and discussed works related to the synthesis of MOFs and MOF-based nanomaterials, and their application as adsorbents in the capture of toxic metal ions and radionuclides. The interaction mechanism was also compared and discussed from the results of advanced spectroscopy measurements and computational theoretical calculations. The MOFs and MOF-based nanomaterials showed rapid sorption kinetics, high sorption ability, superior sorption selectivity, and excellent reusability. The highly ordered porous structures could be controlled during the synthesis processes, thus facilitating the diffusion of metal ions to the inner structure of the MOFs. The grafting of special functional groups to change their surface properties and to modify the porosity or pores of MOFs is an efficient strategy for increasing the sorption capacity and selectivity of target metal ions or radionuclides. On the basis of the above-mentioned studies, it can be concluded that the modification of nanomaterials to increase the physicochemical interaction between radionuclides and nanomaterials could be a suitable method for increasing the sorption ability, for purposes such as: (1) to functionalize metal nodes and organic linkers in nanomaterials; (2) to introduce defects into the structures of nanomaterials; (3) to construct nanomaterial-based composites, etc. [21].

In this work, we summarize and discuss recent results about the application of COFs, MOFs, COF-based materials, MOF-based materials and other carbon-based materials in the efficient removal of radionuclides in the treatment of radioactive environmental pollution. The effects of the structures of the materials and the surface functional groups for highly selective sorption of radionuclides are summarized, and the mechanisms are discussed on the basis of spectroscopy measurements and computational calculations. Finally, the future challenges are discussed.

## 2. Synthesis of Materials

The synthesis of COFs, MOFs and other kinds of carbon-based nanomaterials such as carbon nanotubes and graphene have been summarized in many recent reviews [20,21,22,23]. Generally, the sorption abilities of nanomaterials are influenced by their structures, surface areas, pore size distribution, functional surface groups and hydrophilicity, etc., [24]. The nanomaterials are generally synthesized using the bottom-up blending method and post-surface modification. Such approaches have been widely applied for designing functional nanomaterials, and have been reviewed in detail in many review articles. Herein, we only describe the techniques very simply.

### 2.1. Bottom-Up Technique

The bottom-up technique is a straightforward method, but is a little more difficult for the synthesis of functional nanomaterials, as this method requires the direct building of units with functional moieties. The method has the advantage of homogeneous distribution of surface groups, and thereby improved physicochemical stability. The surface functional groups are helpful for post-modification of nanomaterials to improve the sorption capacities.

### 2.2. Blending Method

The blending technique is efficient for combining functional groups with nanomaterials and therefore elegantly increasing the multiple functions of the different components and improving the physicochemical properties of materials. After the combination of different kinds of functional groups, the stability, sorption capacity and selectivity of nanomaterials could be obviously enhanced. The blending of nanomaterials with other materials can overcome the shortcomings of nanomaterials and extend the applications into multidisciplinary areas.

### 2.3. Post-Surface Modification Method

The post-surface modification strategy is widely applied for the surface grafting of functional groups. In the post-synthesis process, the functional groups can be grafted onto the surface or network of nanomaterials. The free functional groups in the skeleton can act as reaction anchors with other groups through chemical reactions such as enolketo tautomerism, N-H bond deprotonation, imine-amide transformation, etc. Post-surface modification strategies require anchor sites on the nanomaterials for post-synthesis grafting. The technique is an efficient, facile, and practical surface modification method for improving the sorption capacity, as the grafted functional groups can provide more active sites and may also affect the crystal structures of nanomaterials.

## 3. Efficient Elimination of U(VI)

Uranium is one of the most important strategic resources for nuclear power plants. Based on the uranium consumption rate in nuclear power plants, the current storage of uranium (~6.1 Mt) will be used up in about 80–90 years [25]. Thereby, it is imperative to separate or recover U(VI) from different sources, such as nuclear wastewater, spent fuel, salt lakes, oceans, etc. Wang’s group [26] and Lin’s group [27] first applied MOFs in the removal of U(VI) from aqueous solutions. Different kinds of functional groups, i.e., amino, amidoxime, amide, carboxyl, hydroxyl, phosphono, and imidazole groups, were introduced into the frameworks of MOFs as binding sites to improve the sorption of U(VI) [28]. Wang et al. [29] prepared Zn-based MOFs, which showed a maximum sorption capacity of 125 mg/g with ultrafast sorption kinetics (1 min to reach equilibrium) at pH = 2. The carboxyl groups in the frameworks contributed to the uptake of U(VI). Chen et al. [30] synthesized amidoxime-functionalized MOF (denoted as UiO-66-AO) using a post-synthetic modification technique, which showed relatively high selective adsorption ability of 2.68 mg/g U(VI) from real seawater, and the extended X-ray absorption fine structure (EXAFS) spectroscopy analysis suggested that the hexagonal bipyramid coordination of U(VI) with the amidoxime ligands contributed to the binding of U(VI) to UiO-66-AO. Wang et al. [31] synthesized a hierarchical-porous Cu-MOF with ordered macropores (100–200 nm, Figure 1a,b), which had a micro–macro hierarchical architecture. The Cu-MOF showed high sorption capacity for U(VI) (563 mg/g at pH 5.5) and two-step sorption kinetic processes (Figure 1c) from wastewater. The fast kinetic process (within 2 h to achieve equilibrium) was mainly dominated by surface binding of U(VI) in the macropores, whereas the slow kinetic process was attributed to the low mass transfer in the micropores. The high sorption of U(VI) on Cu-MOF was attributed to the cooperative effect of U(VI) interaction with the binding sites in the micro- and macropores. Yuan et al. [32] synthesized different kinds of UiO-66-derived MOFs and applied them for the extraction of U(VI) from real seawater. A high sorption capacity of 6.85 mg/g from real seawater was achieved, which was mainly attributed to the carboxyl and amino functional groups in the MOF skeleton. The density functional theory (DFT) calculations showed the highest binding energy (−2.29 eV) of U(VI) to be with UiO-44-3C4N, except for UiO-66-3C5C (Figure 2a). Two oxygen atoms of the carboxyl groups coordinated with the U(VI), with bond distances of 2.27 and 2.04 Å. Although the UiO-66-3C5C had higher binding energy (−2.51 eV) than UiO-44-3C4N (−2.29 eV), the nanopocket formed in UiO-66-3C5C was smaller than that in UiO-66-3C5C. The smaller structure of UiO-44-3C4N provided higher binding ability to U(VI) ions, meaning that the structure restricted the entrance of other kinds of metal ions with sizes larger than U(VI) ions. The XAFS analysis of U(VI)-loaded UiO-44-3C4N further indicates that the oxygen atoms in the carboxyl groups bind U(VI) at equatorial plane and amino groups form hydrogen bonds (Figure 2b). Two axial O and two coordination O interactions were detected with U(VI), and the bond distances of U = O and U-O were in good agreement with the results of DFT the calculations, which were performed using the Gaussian 09 software package.

Cheng et al. [33] synthesized polyarylether-based COFs with chain amidoxime groups (denoted as COF-HHTF-AO) and used for U(VI) selective extraction from real seawater. The sorption capacity reached up to 5.12 mg/g, which was 1.61 times higher than that of V(V). It is well known that the presence of V(V) in natural seawater is the most important parameter for the competition extraction of U(VI) to nanomaterials. High selectivity for U(VI) has never been found in the COF-based nanoadsorbents reported to date. The interaction of U(VI) and V(V) with COF-HHTF-AO was further simulated by DFT calculation using the Gaussian 09 software package (Figure 3a). The Becke three-parameter Lee Yang Parr function was used to evaluate the geometries and energies. The 6–31 G(d) basis set was applied to describe the H, C, N, O atoms, while the U and V atoms were treated using the Stuttgart–Dresden–Bonn basis set. It is well known that U(VI) mainly exists as uranyl tricarbonate species ([UO_2_(CO_3_)_3_]^4−^) in natural seawater [34], where the amidoxime groups compete with CO_3_^2-^ to bind with [UO_2_(CO_3_)_3_]^4−^ species [32]. The amidoxime group coordinates with oxime oxygen and nitrogen in [UO_2_(CO_3_)_3_]^4−^. The six-coordinate structure (COF-UO_2_-(CO_3_)_2_) of U(VI) binding with amidoxime groups and two carbonate groups gave the highest adsorption energy of −19.38 kcal/mol, suggesting the most stable structure of UO_2_-CO_3_ complexes with COF-HHTF-AO. The binding of V(V) with COF-HHTF-AO (Figure 3b) showed that HVO_4_^2-^ and H_2_VO_4_^-^ interacted with amidoxime through hydroxyl in amidoxime or hydrogen bonding with amine. The adsorption energy of V(V) with COF-HHTF-AO is much lower than U(VI), suggesting the preferential interaction of COF-HHTF-AO with U(VI). The charge transfer density of U(VI) and V(V) (Figure 3c) showed that the transferred charge of U(VI) (0.946 a.u.) was higher than those of HVO_4_^2-^ (0.835 a.u.) and H_2_VO_4_^-^ (0.818 a.u.). The DFT calculation suggested the selective binding of COF-HHTF-AO for U(VI) ions rather than V(V) ions. Wang’s group from Hainan University [35,36,37,38,39] extensively studied the high selective extraction of U(VI) from real seawater using different kinds of organic framework-based nanomaterials, and achieved significant results. Their outstanding work showed that their materials could find possible application for the selective preconcentration/extraction of U(VI) from seawater in the South Sea of China.

The photocatalytic technique is a promising method for U(VI) photoreduction. The tetravalent U(IV) has low solubility and low mobility in the natural environment. Therefore, the reduction of hexavalent U(VI) to tetravalent U(IV) under complicated conditions, especially at extra-low concentrations, is an efficient method for achieving U(VI) fixation [40,41]. Under visible light irradiation, the photocatalyst is photo-excited to generate electron–hole pairs, which can reduce the surface-adsorbed hexavalent U(VI) to pentavalent U(V) or insoluble tetravalent U(IV), and thereby immobilize U(VI) on solid surfaces [42]. In our previous work, we [43] investigated the effect of Cr(VI) and bisphenol A (BPA) on the adsorption/photocatalytic reduction of U(VI) by C_3_N_4_ under visible light conditions. U(VI) could be photocatalytically reduced to U(VI) to form inner-sphere complexation and surface precipitation. In the X-ray absorption near edge structure (XANES) spectra, the sorption-edge energies of U(VI)-adsorbed C_3_N_4_ after irradiation times of 30 min (~17,176.6 eV), 60 min (~17,176.6 eV) and 240 min (~17,176.5 eV) are very close to those of standard U(IV) samples (~17,176.2 eV), but much different to that of U(VI) samples (~17,178.5 eV) (Figure 4A). The XANES spectra provide evidence of the reduction of U(VI) to U(IV) on the C_3_N_4_ surface after visible light irradiation. The Fourier transform peaks of the U(VI)-adsorbed C_3_N_4_ showed the formation of U-O_ax_ and U-O_eq_ bonds, and another peak of U-U shell suggested the formation of precipitation. The EXAFS Fourier transform (FT) spectra could also be fitted by a C-U bond at 3.78Å, suggesting the formation of inner-sphere complexes on the C_3_N_4_ surface (Figure 4B). The presence of BPA and Cr(VI) significantly enhanced the photocatalytic reduction of U(VI). U(VI) was reduced to U(IV) by photogenerated electrons. Cr(VI) was photocatalytically reduced to Cr(III) by H_2_O_2_. The reaction of Cr(VI) and H_2_O_2_ promoted the generation of O_2_^-*^ radicals, which enhanced BPA photocatalytic degradation. Most importantly, BPA and its by-products were able to capture the photoinduced electrons, promoting the separation of holes and electrons, and thereby enhancing the photocatalytic reduction of U(VI) to U(IV) (Figure 4C). The presence of other kinds of metal ions and organic pollutants could improve the adsorption ability of the materials, increasing the photoreduction–solidification of U(VI), which is crucial for the immobilization/extraction of U(VI) from very complicated wastewater if the structures of the nanomaterials are well constructed [44].

Our group [45] reported a new strategy for the extraction of uranium from natural seawater by converging the cooperative functions of adsorption−photocatalysis into the space of covalent organic frameworks (COFs). As shown in Figure 5a, the presence of amidoxime groups in the COFs offered selective binding sites for uranyl, while triazine units and bipyridine−Pd groups acted cooperatively to reduce adsorbed U(VI) to a U(IV) solid product for facile collection. One of our developed COFs, named 4−Pd−AO, displayed exceptional performance in sequestering and reducing uranyl from spiked seawater, with a removal efficiency of >98.6% under visible light irradiation (Figure 5b). The XANES spectra in Figure 5c demonstrate that 4−Pd−AO reduced the adsorbed UO_2_^2+^ to UO_2_. Excellent antibacterial and antialgal activities were observed for sustained efficient uranium extraction performance (Figure 5d). This study established multicomponent COFs as promising candidates for efficient U(VI) extraction from natural seawater.

Yang et al. [46] demonstrated a sorption-electrocatalytic technology for the effective capture of U(VI) from real seawater using a Fe−N_x_−C−R catalyst. The amidoxime groups offered the hydrophilicity to selectively capture U(VI), while the FeN_x_ single-atom centers reduced the surface-adsorbed UO_2_^2+^ to UO_2_^+^. Most importantly, the unstable UO_2_^+^ was re-oxidized to U(VI) in real seawater to form solid Na_2_O(UO_3_·H_2_O)_x_ through electrodeposition, which could be easily extracted from the seawater (Figure 6). The water contact angle (40^°^) of Fe−N_x_−C−R demonstrated the high hydrophilicity (Figure 6a). The sorption capacity of Fe−Nx−C−R (Figure 6b) was 129.9 mg/g, and the kinetic sorption (Figure 6c) was well simulated by the pseudo-second-order kinetic model, suggesting a chemical sorption process [47]. The EXAFS spectrum of the U(VI)-loaded sample is allied to that of UO_2_(NO_3_)_2_·6H_2_O, suggesting the formation of U(VI)O_2_^2+^ (Figure 6d). In cyclic voltammetry (CV) tests (Figure 6f), a sharp peak was found at −0.77 V (vs. SCE), which corresponded to the electroreduction of U(VI) to U(V). The peak at −0.43 V (vs. SCE) corresponded to U(V) re-oxidation to U(VI) [48,49]. The extraction of U(VI) from 10 ppm U(VI) spiked real seawater using the Fe−N_x_−C−R as sorbent−electrocatalyst showed a maximum extraction capacity of 128 mg/g after 1500 min (Figure 6g). The Fe−N_x_−C−R/carbon felt electrode was applied to extract U(VI) from 1000 ppm U(VI)-spiked real seawater (Figure 6h). A yellow product was formed on the electrode, suggesting the preconcentration of U(VI) ions from natural seawater on the electrode.

The adsorption–electrocatalytic reduction mechanism of U(VI) by Fe−N_x_−C−R is shown in Figure 7a. The FeN_x_ sites (initially as Fe(II)N_4_ center) reduced the surface-adsorbed U(VI)O_2_^2+^ to U(V)O_2_^+^ [48,50]. Then, the Na_2_O(UO_3_·H_2_O)_x_ was formed by U(V) re-oxidation to U(VI) in the presence of Na^+^ ions. The unstable U(V)O_2_^+^ transferred electrons to the Fe(III)N_4_ center, and regenerated Fe(II)N_4_. The reversible electron transfer between the FeN sites and U(VI)O_2_^2+^/U(V)O_2_^+^ resulted in the formation of Na_2_O(UO_3_·H_2_O)_x_ precipitation in the electrocatalysis reduction-oxidation processes in the presence of Na^+^ ions. Li et al. [51] modified MOFs with phosphono and amino groups (denoted as PN-PCN-222), and applied for the photocatalytic reduction of U(VI) from wastewater. The PN-PCN-222 showed high selective capture of U(VI) in the presence of non-redox-active competing metal ions. U(VI) was reduced to U(V), which was further automatically disproportionated into U(IV) and U(VI). The formation of (UO_2_)O_2_·2H_2_O was attributed to the reaction of H_2_O_2_ with UO_2_. The electrons reacted with uranyl to form UO_2_ and UO_2_^+^. The UO_2_^+^ was further disproportionated to U(VI) and U(IV), where U(VI) reacted with H_2_O_2_ to form (UO_2_)O_2_·2H_2_O. The photocatalytic reduction of U(VI) to U(IV) facilitated the extraction of U(VI) from the wastewater (Figure 7b). This strategy overcomes the drawbacks of the sorption process to improve the capability of photoactive MOFs. Zhang et al. [52] synthesized SnO_2_/CdCO_3_/CdS nanocomposites and applied for the photocatalytic reduction extraction of U(VI) from wastewater. The results showed high removal of U(VI) in a short irradiation time in the presence of different heavy metal ions. In the wavelength range of 400–520 nm, the SnO_2_/CdCO_3_/CdS showed very high performance in the photoreduction of U(VI). The light absorption from the transition between inter-band defects and inter-band contributed to U(VI) reduction. The energy level of SnO_2_/CdCO_3_/CdS was helpful for the separation of holes and photoelectrons, and the accumulation of photo-generated holes on CdS and electrons on SnO_2_, which promoted the photocatalytic reduction of U(VI) to U(IV). CdS offered photoelectrons and holes, CdCO_3_ enhanced the separation of photoelectrons and holes. SnO_2_ stabilized the CdS against photo-corrosion. They provided a simple and economical photocatalyst for the effective efficient extraction of U(VI) from wastewater in possible real applications. Cai et al. [53] firstly reported the piezo catalytic reduction–oxidation strategy for the effective extraction of U(VI) from complicated solutions using Zn_2_SnO_4_/SnO_2_ as piezo catalyst. The adsorbed U(VI) could be reduced to UO_2_ by piezo electrons, and then re-oxidized to form (UO_2_)O_2_∙2H_2_O, which was efficiently separated from the complicated solutions. The Zn_2_SnO_4_/SnO_2_ showed high selectivity and recycling ability. Li et al. [54] found an eco-friendly method for extracting U(VI) with high selectivity from wastewater under visible light conditions without a solid catalyst. In the presence of alcohol, the efficient ligand-to-metal charge transfer with uranyl complexes could reduce U(VI) to U(V) under visible light conditions in the pH range of 5–6.5, and then U(V) was transformed to U(VI) ions and UO_2_ through the disproportionation reaction [55]. This work provided a new facile method for the sensitive extraction of U(VI) from complicated solutions without solid catalyst under solar light conditions, which is crucial to extract U(VI) from wastewater in real applications. By adjusting the solution pH and adding a small amount of alcohol to the solution, the uranium can be separated under natural solar light irradiation. 

## 4. Efficient Elimination of ^99^TcO_4_^-^

^99^TcO_4_^-^, one of the most significant fission products, is problematic because of its long half-life and high mobility in the natural environment. The efficient elimination of ^99^TcO_4_^-^ is crucial for preventing the potential dangers of ^99^TcO_4_^-^ to human health. Wang’s group [56] from Soochow University synthesized two-dimensional conjugated cationic COFs (named as SCU-COF-1) and applied ^99^TcO_4_^-^ removal under extreme conditions. The SCU-COF-1 was able to achieve sorption equilibration in 1 min, with an ultrahigh sorption capacity of 702.4 mg/g and good selectivity. Da et al. [1] synthesized hydrolytically stable cationic COFs and applied a nonradioactive element which has very similar properties with ^99^TcO_4_^-^ for ReO_4_^-^ removal, and found that the hydroxyl neutral edge sites and the chloride anions in the pores of the COFs resulted in a high sorption capacity of 437 mg/g with extremely rapid sorption kinetics at pH values ranging from 3 to 12. More importantly, the material still showed high sorption efficiency in simulated Hanford Law Stream in the presence of other competing ions such as PO_4_^3-^, CO_3_^2-^, NO_3_^-^ and SO_4_^2-^. Hao et al. [57] synthesized pyridinium salt-based cationic COFs (denoted as PS-COF-1) with the largest surface area (2703 m^2^/g), which was the highest value of today’s reported COF materials, and used it for the separation of ReO_4_^−^ and ^99^TcO_4_^−^ from aqueous solutions. The synthesized PS-COF-1 showed high sorption capacity (1262 mg/g for ReO_4_^−^) with fast sorption kinetics and high selectivity in the presence of other anions at high excess concentrations. More importantly, the PS-COF-1 was able to decrease the ^99^TcO_4_^−^ and ReO_4_^−^ concentrations in polluted wastewater (10 ppb) to the level of drinking water (0 ppb) in 10 min with strong sorption affinity and high stability. The computational calculation revealed that the cationic pyridine sites in PS-COF-1 framework had high interaction affinity with ^99^TcO_4_^−^ and ReO_4_^−^, which contributed high selectivity for the removal of ReO_4_^−^ and ^99^TcO_4_^−^ from solutions (Figure 8). 

Other than COF materials, Drout et al. [58] prepared mesoporous Zr MOFs (Zr_6_-MOFs) and used them for the removal of ReO_4_^-^ and ^99^TcO_4_^-^ from aqueous solutions in the presence of different competing anions. The Zr_6_-MOFs showed a sorption capacity of 210 mg/g for ReO_4_^-^ and achieved equilibrium within 5 min. The large pores of Zr_6_-MOFs facilitated the diffusion of radionuclides to the nodes. Nonchelating and chelating ReO_4_^-^ binding structures were present in the mesopores and small pores of Zr_6_-MOFs. Mei et al. [59] synthesized CB8-based cationic supramolecular MOF, denoted as SCP-IHEP-1, through supramolecular collaborative assembly. The tetrahedral pores of CB8 moieties and the anion-adaptive rearrangement of CB8-surrounded pores were reasonable for the capture of ReO_4_^-^ and ^99^TcO_4_^-^ with high selectivity and sorption capacity. The results showed that soft anion-adaptive MOFs were efficient materials for the selective ReO_4_^-^ and ^99^TcO_4_^-^ binding from complicated systems in presence of other competitive anions. Wang’s group from Soochow University [60] synthesized 8-fold interpenetrated 3D cationic MOFs, named SCU-100, with a tetradentate N-donor ligand and two coordinated Ag^+^ ions as open sites. The SCU-100 showed rapid and high selectivity for ^99^TcO_4_^-^ and ReO_4_^-^ in the presence of other competitive anions such as NO_3_^-^, CO_3_^2-^, PO_4_^3-^, SO_4_^2-^. The SCU-100 also had high stability over a large pH range and excellent β/γ radiation resistance. The single-crystal XRD analysis showed that ReO_4_^-^ was coordinated to open Ag^+^ sites to form Re-O-Ag bonds and also formed a series of hydrogen bonds. The highly selective removal was attributed to the selective coordination to the open Ag^+^ sites through the formation of Ag-O-Tc bonds (Figure 9). 

Wang’s group also synthesized a hydrolysis-resistant cationic MOF, which showed excellent ion exchange ability for the binding of ^99^TcO_4_^-^ with fast sorption dynamic and high selectivity. The selective removal efficiency was not obviously affected, even at SO_4_^2-^ concentrations 6000 times higher, and the high selectivity on the binding sites of MOFs was well characterized and identified by advanced spectroscopy analysis at the molecular level [61]. They further synthesized a 2D cationic MOF (named as SCU-103) for ^99^TcO_4_^-^ separation from a Savanah River site, and the SCU-103 showed fast kinetic, high selectivity and superior sorption capacity. The concave–convex layers in SCU-103 provided the selective recognition sites for the binding of ^99^TcO_4_^-^ (Figure 10). The representative anion exchange of ^99^TcO_4_^-^ with NO_3_^-^ was shown in Figure 10a. ^99^TcO_4_^-^ entered into the cavity and NO_3_^-^ was oscillated around its original binding sites. The dynamic interaction of ^99^TcO_4_^-^ with SCU-103 were performed using GROMACS 5.1.4 software applying with OPLS-AA force field [62]. The results of DFT calculation indicated that the binding free energies for ^99^TcO_4_^-^ and NO_3_^-^ with SCU-103 indicated the energetically favorable binding interaction, and the binding affinity of ^99^TcO_4_^-^ was much higher than that of NO_3_^-^ (Figure 10b,c). The binding energy of ^99^TcO_4_^-^ at the most energetically favorable site was −17.9 kJ/mol, which was much higher than that of NO_3_^-^, suggesting the favorable exchange of NO_3_^-^ for ^99^TcO_4_^-^. The results showed that the cationic MOFs were able to remove ^99^TcO_4_^-^ from complicated solutions efficiently and with high sorption selectivity and stability [12].

To date, the elimination of ^99^TcO_4_^-^ from radioactive wastewater has been extensively studied, and the interaction mechanism has been intensively investigated using ReO_4_^-^ as a nonradioactive surrogate. The precipitation method is one of the simplest techniques for separating ^99^TcO_4_^-^ from solution. However, the soluble ^99^TcO_4_^-^ cannot be converted to insoluble Tc(IV) and the Tc(IV) incorporated into bulky materials using most current precipitating agents. Cationic MOFs and COFs are promising nanomaterials for the efficient removal of ^99^TcO_4_^-^ from aqueous solutions with fast dynamics, high selectivity, and high sorption capacity. However, there are still some challenges for the application of MOFs or COFs in the capture of ^99^TcO_4_^-^ from complicated systems. It is still necessary to increase the sorption capacity of COFs or MOFs. Small neutral molecules as linking agents could enhance the sorption ability. The selectivity of ^99^TcO_4_^-^ in the presence of different kinds of anions is still a challenge. The cationic framework and introduction of open metal sites could improve the sorption selectivity. For some cases, the stability of COFs and MOFs is not high enough under extreme conditions, limiting the real application of these materials in the separation of ^99^TcO_4_^-^ from radioactive wastewater.

## 5. Efficient Elimination of Other Radionuclides

Apart from U(VI) and Tc(VII), other kinds of radionuclides can also be efficiently eliminated by nanomaterials. Strontium-90 (^90^Sr) (T_1/2_ = 28.8 a) is a β emitter. ^90^Sr has drawn extensive environmental concern because of its high radiotoxicity and mobility. A 2D-layered anionic polymer (named as SZ-4) was synthesized [63] and applied for high elimination of Sr(Ⅱ) in strong acidic solutions. The SZ-4 showed high sorption capacity (117.9 mg/g), which was attributed to the coordination of Sr(Ⅱ) with fluorine donors and oxygen donors in the SZ-4 layer. Sr(II) was able to be successfully removed from seawater, because the influence of the hard cations Ca(Ⅱ) and Na(Ⅰ) in seawater is very weak. Zhang et al. [64] synthesized carboxyl-functionalized UiO-66 and applied it for the extraction of Th(IV), and the results showed an adsorption capacity of 350 mg/g with high selectivity and rapid kinetics. Lu et al. [65] fabricated carboxyl-functionalized 3D-COOH-COF and applied it for the sorption of Nd^3+^. The results showed the selective adsorption of Nd^3+^ over Sr^2+^ and Fe^3+^, which was attributed to the strong interaction of the carboxyl group with Nd^3+^ ions. Zhong et al. [66] synthesized TpPa-1 COF and applied it for Eu(III) removal. The high sorption capacity (1107.63 mg/g) was attributed to the large amounts of active sites and intramolecular hydrogen bonds on the TpPa-1 COF pore wall. Xiong et al. [67] used [NH_4_]^+^[COF-SO_3_^−^] membrane to separate Th(IV) from solutions containing U(VI), Th(IV), and rare earth elements. The results showed high adsorption capacity for Th(IV) (395 mg/g) with high selectivity. Gao et al. [68] prepared FJMS-InMOF and applied for simultaneous removal of Cs(I) and Sr(Ⅱ). FJMS-InMOF efficiently extracted ppb-level of Cs(I) and Sr(Ⅱ) even with high concentrations of alkaline earth and alkali metal. Liu et al. [69] synthesized COFs using the microwave irradiation method and applied them for radioactive iodine capture. The results showed an excellent sorption capacity of 2.99 g/g for volatile ionide and 493 mg/g for iodine-cyclohexane solution. Yu et al. [70] reviewed the recent work for the elimination of radionuclides such as U(VI), Eu(III), Sr(II), Cs(I) from solution by MXenes materials, and concluded that the interlayer space and functional groups were critical for the uptake of radionuclides. The sorption of U(VI) and Eu(III) was mainly dominated by surface complexation, whereas the sorption of Sr(II) and Cs(I) was mainly attributed to ion exchange. Cheng et al. [71] synthesized a co-doped MOF and graphene oxide composite using an electrostatic self-assembly process and applied it for the removal of Cs^+^ from aqueous solutions, and the results showed high sorption capacity, which was dominated by chemisorption and electrostatic interaction between Cs^+^ and composite functional groups. From the above-mentioned results, it can be seen that the surface modification of nanomaterials with special functional groups could increase the sorption ability and selectivity, which is crucial for the special applications in the extraction of target radionuclides from complicated solutions. It is also necessary to note that most kinds of the nanomaterials are stable in strong acid or base solutions, and against radioactive irradiation, making them a possibility for real applications.

## 6. Conclusions and Perspective

In this review article, the application of different kinds of nanomaterials, especially COFs and MOFs, was reviewed for the removal of different radionuclides such as U(VI), Tc(VII) and other kinds of radionuclides. The sorption of radionuclides, especially the selective extraction of target radionuclide was discussed in detail. Improving the selective sorption of the target radionuclides from complicated solutions, especially from real seawater, is still the main challenge. Surface modification with special functional groups to form strong complexes with target radionuclides such as U(VI) or Tc(VII) is a possible strategy. The stability and reusability of the nanomaterials should be considered. Sorption–reduction–solidification can potentially increase the sorption ability of nanomaterials, and thereby efficiently extract the radionuclides.

The efficient extraction of U(VI) from seawater is still a challenge for the future because of its low concentration in natural seawater (~3.3 ppb U(VI)) and the presence of high concentrations of competing ions such as V(V) ions. It is well known that there are about 4.5 billion tons of U(VI) in seawater, which is sufficient to ensure nuclear energy utilization for long-term development. For highly selective enrichment of U(VI) from natural seawater, the sorption capacity, rapid kinetics, selectivity, and especially the antibacterial ability should also be considered.

## Figures and Tables

**Figure 1 nanomaterials-12-01443-f001:**
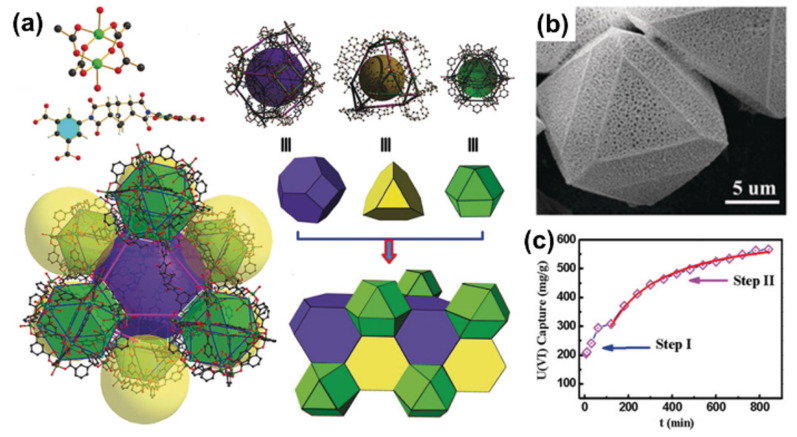
Structure of **USC-CP-1** (the purple, yellow and green units represent three different micropores), (**a**) molecule used for synthesis of **USC-CP-1**, (**b**) SEM of **USC-CP-1**, (**c**) equilibrium data with continuous line fit based on the Langmuir model of uranyl adsorption on **USC-CP-1** at pH 5.5 [31], Copyright 2019 John Wiley and Sons.

**Figure 2 nanomaterials-12-01443-f002:**
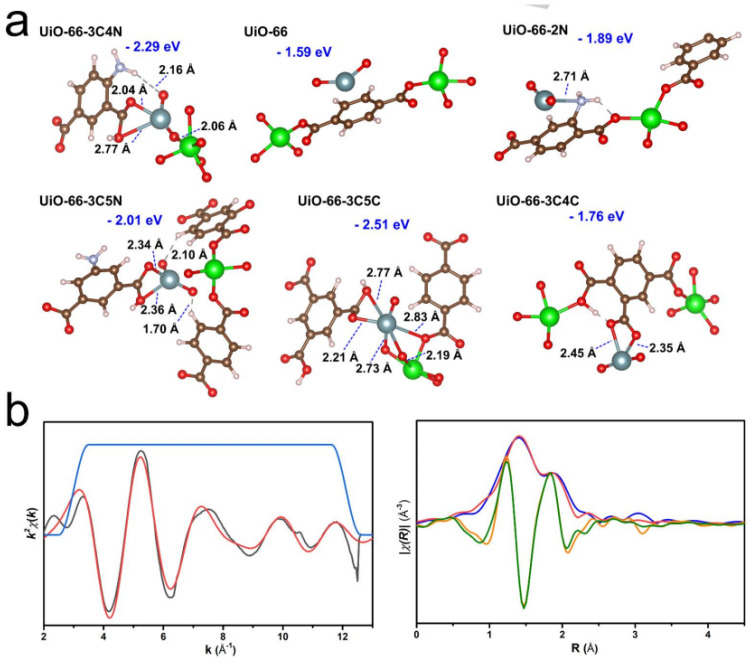
Analysis of the coordination mechanism. (**a**) DFT calculation of the binding energy. The binding energy and the bond distances are shown, (**b**) EXAFS data and fitting results. For the *k*^2^-weighted χ(k) result, the black line and red line indicate the experimental and the fitting data, respectively. For the R space result, the blue line and the red line indicate the Fourier-transformed *k*^2^-weighted experimental and fitting data, respectively. The orange line and green line indicate the real part component and the fitting data, respectively [32], Copyright 2020 John Wiley and Sons.

**Figure 3 nanomaterials-12-01443-f003:**
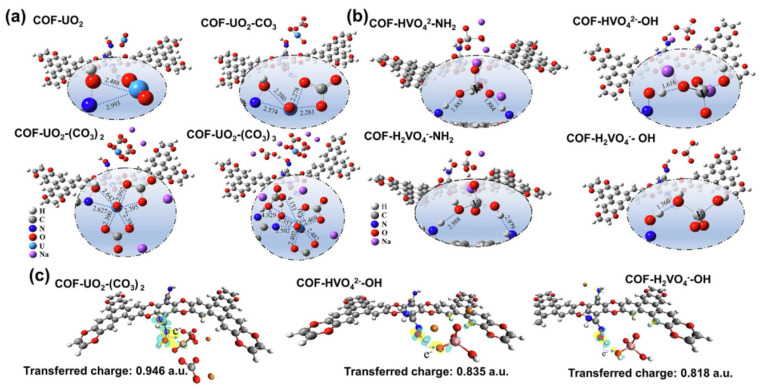
Interaction configuration and binding mechanism between U(VI) or V(V) and COF-HHTF-AO based on DFT simulation. The optimized interaction structures of COF-HHTF-AO with uranyl in the presence of different carbonate contents (**a**) and two forms of vanadium oxide (HVO_4_^2-^ and H_2_VO_4_^-^) (**b**), which are neutralized by Na ions; bond lengths are in Å, (**c**) Charge density difference of the stable configuration of uranyl, HVO_4_^2-^, and H_2_VO_4_^-^ adsorbed on COF-HHTF-AO. The yellow and cyan regions indicate increases and decreases in charge density, respectively [33], Copyright 2021 Elsevier.

**Figure 4 nanomaterials-12-01443-f004:**
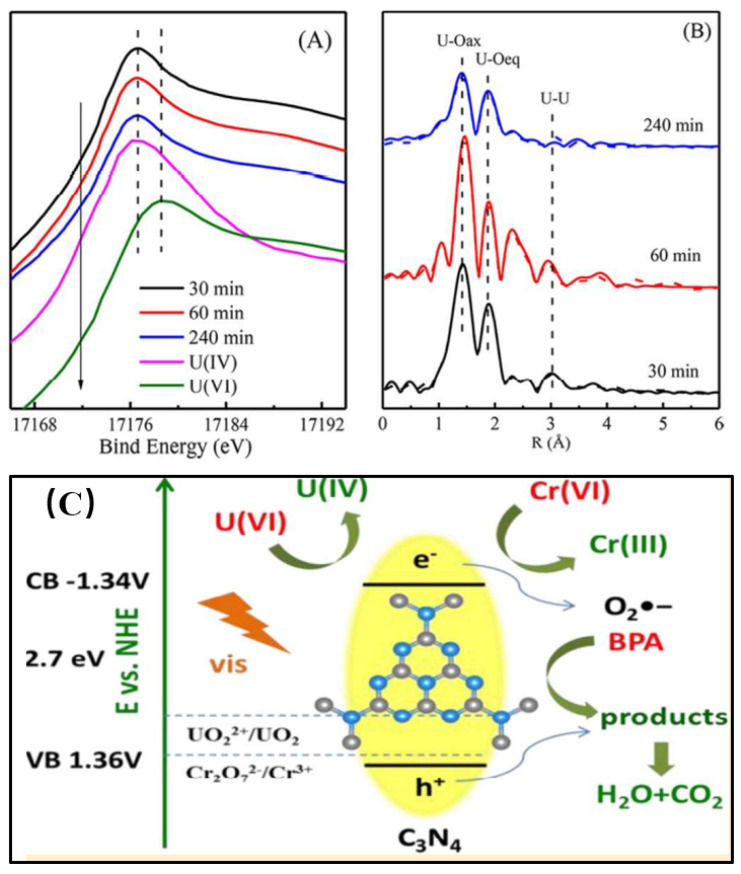
XANES (**A**) and EXAFS (**B**) analysis of U(VI)-loading C_3_N_4_ at different reaction times. Schematic illustration of photocatalytic reduction of U(VI) and Cr(VI) (**C**) [43], Copyright 2019 American Chemical Society.

**Figure 5 nanomaterials-12-01443-f005:**
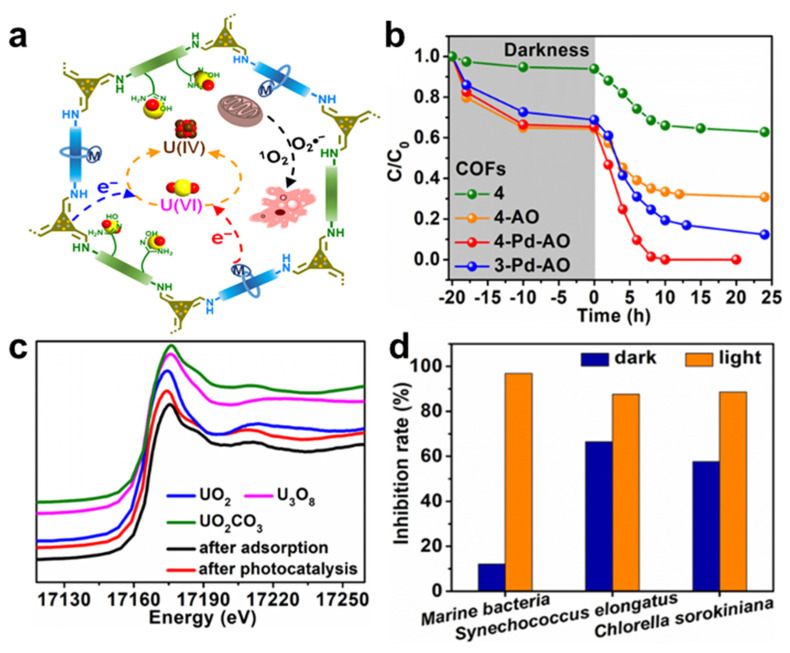
(**a**) Adsorption−photocatalytic strategy for uranium extraction, (**b**) uranium extraction from spiked seawater using adsorbent−photocatalysts, (**c**) U L_III_-edge XANES spectra for COF 4-Pd-AO after uranium extraction studies, and (**d**) antibiofouling spectrum of COF 4-Pd-AO [45], Copyright 2022 Chinese Chemical Society.

**Figure 6 nanomaterials-12-01443-f006:**
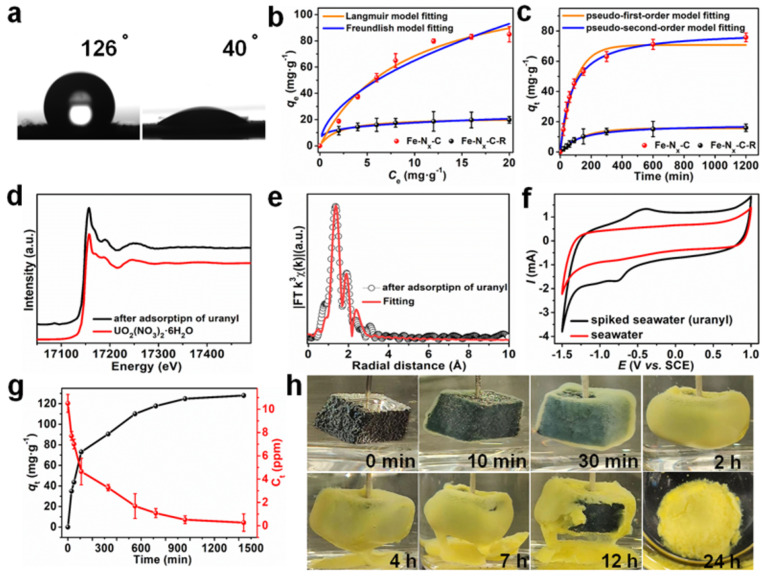
(**a**) Contact angles for deionized water on pressed pellets of Fe-N_x_-C (left) and Fe-N_x_-C-R (right), (**b**) Equilibrium adsorption isotherms for uranyl ion adsorption on different materials at a fixed material-to-solution ratio of 0.1 mg·mL^−1^ in uranyl-spiked seawater (from 0 to 20 ppm), (**c**) Uranyl ion adsorption kinetics on different materials at an initial UO_2_^2+^ concentration of 10 ppm in uranyl−spiked seawater, (**d**) U L_III_-edge XANES spectra for Fe-N_x_-C-R after adsorption of uranyl, and UO_2_(NO_3_)_2_·6H_2_O, (**e**) U L_III_-edge EXAFS R-space and corresponding fitting curves for Fe-N_x_-C-R after adsorption of uranyl, (**f**) Cyclic voltammograms for uranyl-spiked seawater and natural seawater, (**g**) Uranium extraction from spiked seawater with initial uranium concentrations of ~10 ppm, using Fe-N_x_-C-R as an adsorbent−electrocatalyst, (**h**) Photographs of the Fe-N_x_-C-R electrode in spiked seawater (initial uranium concentration of ~1000 ppm) during electrocatalytic extraction [46], Copyright 2021 John Wiley and Sons.

**Figure 7 nanomaterials-12-01443-f007:**
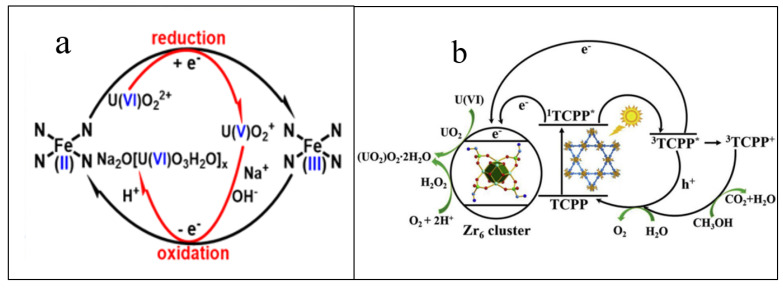
(**a**) Plausible electrocatalytic mechanism for Fe-N_x_-C-R-catalyzed extraction of U(VI) from seawater [46], Copyright 2021 John Wiley and Sons, (**b**) Schematic illustration of selective enrichment and photocatalytic reduction of U(VI) based on PN-PCN-222 [51], Copyright 2019 Elsevier.

**Figure 8 nanomaterials-12-01443-f008:**
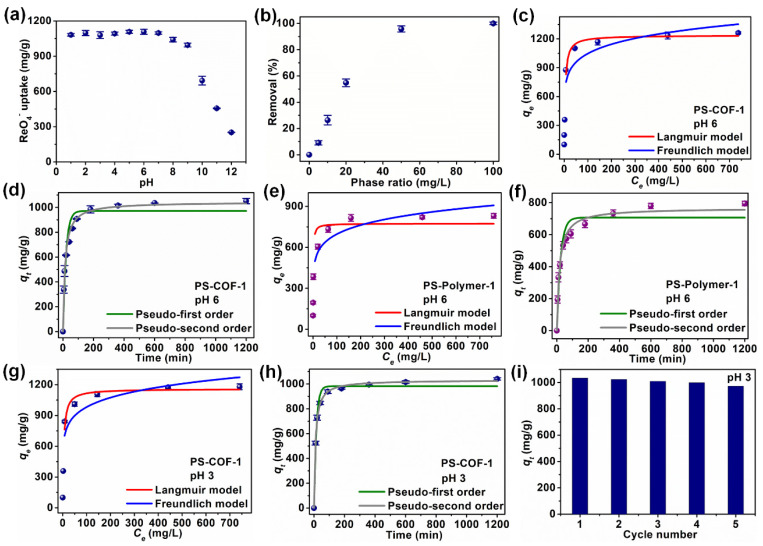
ReO_4_^−^ adsorption results. (**a**) Effect of pH on the removal of ReO_4_^−^, (**b**) ReO_4_^−^ removal by PS-COF-1 at various adsorbent−liquid ratios, (**c**,**e**) Equilibrium adsorption isotherms for ReO_4_^−^ on PS-COF-1 and PS-Polymer-1 at pH 6, respectively, (**d**,**f**) ReO_4_^−^ adsorption kinetics on PS-COF-1 and PS-Polymer-1 at an initial ReO_4_^−^ concentration of ~56 ppm (pH 6), respectively, (**g**,**h**) Equilibrium isotherms and adsorption kinetics of ReO_4_^−^ on PS-COF-1 at pH 3, respectively, (**i**) Recycle test data for ReO_4_^−^removal in HNO_3_ (at pH 3) solutions by PS-COF-1 [57], Copyright 2022 Elsevier.

**Figure 9 nanomaterials-12-01443-f009:**
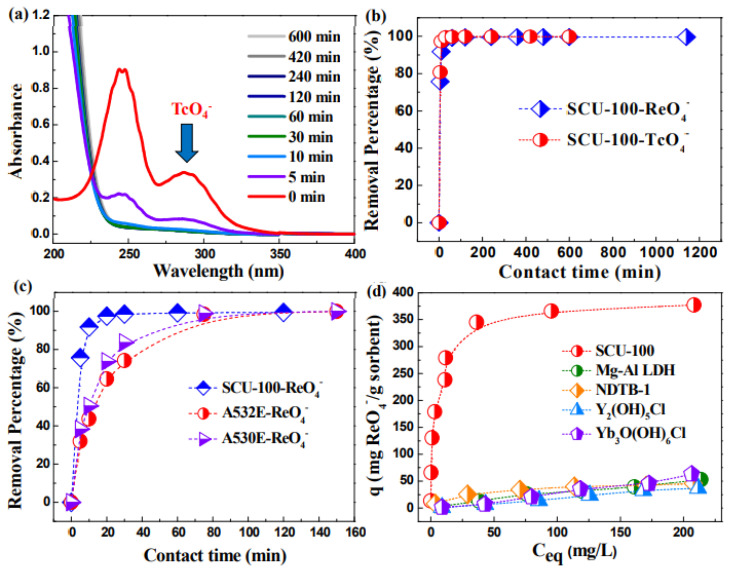
(**a**) UV-vis absorption spectra of aqueous TcO_4_^-^ solution during the anion exchange with SCU-100, (**b**) Removal percentage of TcO_4_^-^ and ReO_4_^-^ by SCU-100 as a function of contact time, (**c**) Comparison of the sorption kinetics of ReO_4_^-^ by SCU-100, A532E, and A530E, (**d**) Sorption isotherms of ReO_4_^-^ by cationic SCU-100 compared with other materials [60], Copyright 2017 American Chemical Society.

**Figure 10 nanomaterials-12-01443-f010:**
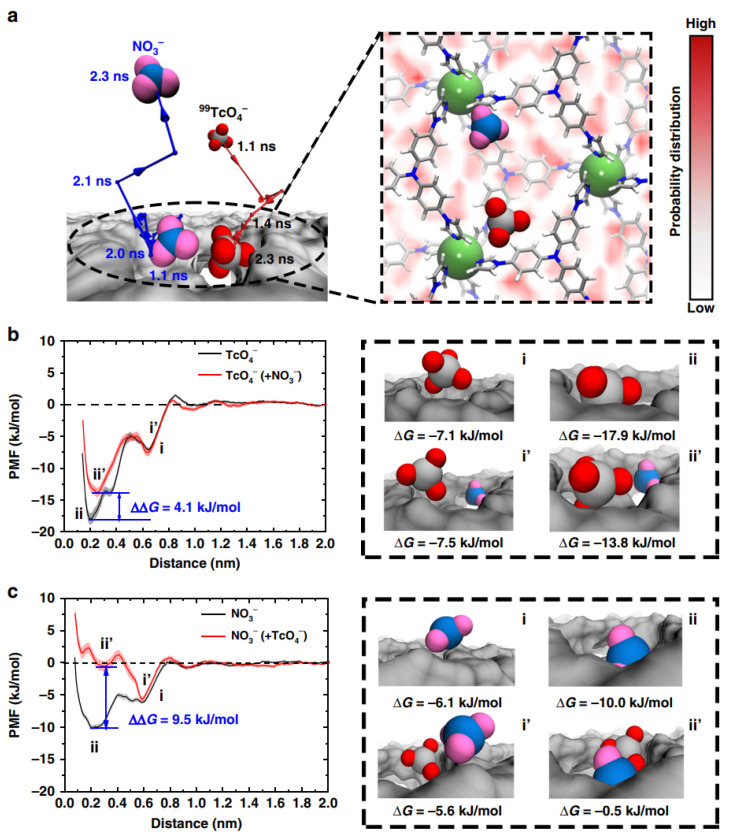
Binding free energies of ^99^TcO_4_^-^ and NO_3_^-^ to their most energetically favorable sites in SCU-103. (**a**) A representative anion-exchange process of NO_3_^-^ by ^99^TcO_4_^-^ (left). The red and blue lines with arrows represent the center of mass motion trajectory of ^99^TcO_4_^-^ and NO_3_^-^ from 1.1 ns to 2.3 ns. The most important crucial intermediate state during the anion-exchange process is shown when ^99^TcO_4_^-^ binds to its final binding site, while NO_3_^-^ dwells at its original binding site (right). The color gradient of the background demonstrates the binding probability distribution of NO_3_^-^ to SCU-103, and ranges from the lowest probability (0, white) to the highest probability (0.110, red). (**b**,**c**) Binding free energies of ^99^TcO_4_^-^/NO_3_^-^ alone to its final binding site (black curves), and the binding free energies of the two anions at the crucial intermediate state (red curves). Snapshots highlighted by the black dashed lines represent the typical binding configurations of ^99^TcO_4_^-^/NO_3_^-^ anions to SCU-103 corresponding to the energy minima in the free energy curves. The shadows in (**b**,**c**) represent the predictive error of the binding free energy profiles [12], Copyright 2020 Springer Nature.

## Data Availability

No new data were created or analyzed in this study. Data sharing is not applicable to this article.

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
