# Peer review of "Efficient Selective Removal of Radionuclides by Sorption and Catalytic Reduction Using Nanomaterials"

_nanomaterials, 2022, doi:10.3390/nano12091443_

Round 1

Reviewer 1 Report

In my opinion, the topic requires the preparation of a good review paper, as there are already several reviews written in recent years. This publication is not very detailed and requires a lot of time form authors to obtain a satisfactory level. For me, this is just a draft of the publication.

For an incomprehensible reason, the authors used some geographic key in the selection of the cited articles.

I am convinced that the work in this form is not suitable for publication.

For an incomprehensible reason, the authors used some geographic key in the selection of the cited articles.

Moreover, in such chapter "2. Synthesis of Materials"

provide at least a few good reviews describing the appropriate syntheses.

Author Response

In my opinion, the topic requires the preparation of a good review paper, as there are already several reviews written in recent years. This publication is not very detailed and requires a lot of time form authors to obtain a satisfactory level. For me, this is just a draft of the publication.

For an incomprehensible reason, the authors used some geographic key in the selection of the cited articles.

Reply: Thank you very much for your kind comments. We have added more references from different countries and institutes in the revised form.

I am convinced that the work in this form is not suitable for publication.

Reply: As a minor review, we think that this manuscript is enough as we just want to give some information in this area.

For an incomprehensible reason, the authors used some geographic key in the selection of the cited articles.

Reply: Thank you very much for your kind comments. We have added more references from different countries and institutes in the revised form.

Moreover, in such chapter "2. Synthesis of Materials" provide at least a few good reviews describing the appropriate syntheses.

Reply: The synthesis of materials have been reviewed in many previous review articles, we just gave some reviews in this manuscript. Thereby we did not describe the synthesis of materials in detail in this review.

Reviewer 2 Report

The article describes an efficient selective removal of radionuclides by sorption and catalytic reduction using nanomaterials. It is a very important study, especially in the green engineering field.

The researchers used interesting techniques for nanomaterials synthesis and DFT calculations.

I think that the authors have to write more about the conditions of DFT calculations.

Also, details about the software which was used for the DFT calculations are missing.

Author Response

The article describes an efficient selective removal of radionuclides by sorption and catalytic reduction using nanomaterials. It is a very important study, especially in the green engineering field.

The researchers used interesting techniques for nanomaterials synthesis and DFT calculations.

I think that the authors have to write more about the conditions of DFT calculations.

Reply: Thanks a lot for the favorable comments. We have more information about DFT calculation in the revised form, and the revised places are marked with Red.

Also, details about the software which was used for the DFT calculations are missing.

Reply: Information about the software used for DFT calculation was different for different references. As a review article, we think that the results of DFT calculation is more important. We also added the software used for DFT calculation from different references in revised form.

Reviewer 3 Report

In this review the authors address a sensitive topic, that of selective sorption or catalytic reduction of target radionuclides using different kinds of nanomaterials such as carbon-based nanomaterials, metal organic frameworks, covalent organic frameworks, and the interaction mechanisms. After a well documented introduction, authors discuss in distinct and extended chapters the removal of U(VI), 99TCO4- and in a short chapter about other radionuclides.

This review can be accepted for publication after some concern and corrections:

Please confirm that for reproduction of figures, authors have permission from publishers.

Some abbreviations are not explained.

In my opinion chapter 4 must be removed. It is not with the stated scope of this review.

Author Response

In this review the authors address a sensitive topic, that of selective sorption or catalytic reduction of target radionuclides using different kinds of nanomaterials such as carbon-based nanomaterials, metal organic frameworks, covalent organic frameworks, and the interaction mechanisms. After a well documented introduction, authors discuss in distinct and extended chapters the removal of U(VI)99TCO4- and in a short chapter about other radionuclides.

This review can be accepted for publication after some concern and corrections:

Please confirm that for reproduction of figures, authors have permission from publishers.

Reply: Thanks for your suggestion. We have got the permission from the publishers.

Some abbreviations are not explained.

Reply: The abbreviations in the main text are given the full expression for the first time in the main text.

In my opinion chapter 4 must be removed. It is not with the stated scope of this review.

Reply: Thank you for your favorable comment. Although this review mainly focused on U(VI) and Tc(VII) removal by nanomaterials, we think it is better to add some information about the elimination of other radionuclides as this is helpful for readers to understand some recent works in this area. As a review, we think a simple description is still important. Thereby we just described the works in this area very simple.

Round 2

Reviewer 1 Report

The review is still missing important items, and still gives the impression of incomplete. 

Examples of works that would be worth considering: 

Bioresource Technology Volume 160, May 2014, Pages 142-149
Environmental Science & Technology 50.8 (2016): 4459-4467
Chemical Engineering Journal Volume 353, 1 December 2018, Pages 157-166
and many others.

I still believe that the authors prepared this work not very thoroughly and did not spend too much time describing the entire scope of the research.

Reviewer 3 Report

Accept.